# Nutritional Issues in Children with Dysphagia

**DOI:** 10.3390/nu16111590

**Published:** 2024-05-23

**Authors:** Pilar Ortiz Pérez, Inés Valero-Arredondo, Encarnación Torcuato-Rubio, Marta Herrador-López, Rafael Martín-Masot, Víctor Manuel Navas-López

**Affiliations:** 1Pediatric Gastroenterology and Nutrition Unit, Hospital Regional Universitario de Málaga, 29011 Málaga, Spain; portizp@gmail.com (P.O.P.); inesvaleroarredondo@gmail.com (I.V.-A.); encatr@gmail.com (E.T.-R.); herradorlopezm@gmail.com (M.H.-L.); victor.navas@gmail.com (V.M.N.-L.); 2PhD Program in Biomedicine, Translational Research and New Health Technologies, University of Málaga, 29010 Málaga, Spain; 3Biomedical Research Institute of Málaga and Nanomedicine Platform (IBIMA Plataforma BIONAND), 29590 Málaga, Spain

**Keywords:** deglutition disorders, malnutrition, dysphagia, pediatric rehabilitation program, gastrostomy, nasogastric tube

## Abstract

(1) Background: Pediatric dysphagia presents significant nutritional challenges, often impacting growth and development due to reduced oral intake, increased nutritional needs, and gastrointestinal complications; (2) Methods: This prospective quasi-experimental study assessed 117 children under 14 years old (20 patients were under 1 year old, 80 were aged 1–7 years, and 17 were older than 7 years), diagnosed with swallowing disorders, to analyze their caloric, macro-, and micronutrient intake and identify potential deficiencies. The severity of dysphagia was established using functional oral intake scales, and dietary records were reviewed over a 3-day period; (3) Results: The study revealed that 39.8% of participants did not meet their total energy expenditure (TEE), highlighting a high prevalence of malnutrition among these children. Furthermore, patients using feeding devices exhibited a significantly lower caloric intake, and over half required significantly modified food textures. After individualized speech therapy and nutritional rehabilitation, participants showed significant improvements in caloric intake, with their energy coverage increasing from 958% to 1198% of the daily requirement. Rehabilitation also improved tolerance to a broader range of food textures; (4) Conclusions: This research underscores the importance of multidisciplinary, individualized nutritional strategies to address the specific challenges of pediatric dysphagia, emphasizing the role of enteral nutrition and therapeutic interventions in improving the quality of life and nutritional outcomes of these children. Further studies are recommended to assess the long-term impact of such strategies.

## 1. Introduction

Dysphagia is the difficulty or alteration in swallowing, specifically in the transport of food from the mouth to the stomach. There are two main types of dysphagia: oropharyngeal dysphagia (OPD) and esophageal dysphagia. OPD is a symptom that occurs in the context of a neurological, anatomical, or combined disorder. It leads to increased morbidity and mortality and results in significant healthcare costs, with a prevalence of 0.9% in the pediatric population [1]. In children, OPD can stem from various potential risk factors and underlying pathophysiologic mechanisms. Neurological disorders, such as cerebral palsy, muscular dystrophy, and congenital brain malformations, are significant contributors, as they can impair the nerves and muscles essential for swallowing. Anatomical abnormalities, including cleft palate and tracheoesophageal fistula, can also obstruct the normal passage of food. Premature birth is another risk factor, as premature infants often have underdeveloped swallowing reflexes and coordination issues. Gastroesophageal reflux disease (GERD) and certain genetic syndromes, such as Down syndrome, can further complicate swallowing in pediatric patients. The pathophysiologic mechanisms in pediatric OPD involve disruptions in the intricate coordination of muscles and nerves. In oropharyngeal dysphagia, neurological impairments may result in uncoordinated or weak muscle contractions, leading to difficulty initiating swallowing and an increased risk of aspiration [1,2,3].

OPD presents a significant challenge in pediatric populations, affecting not only the child’s ability to consume a variety of foods but also their nutritional status, growth, and overall development [4]. Literature reviews emphasize the importance of early diagnosis of dysphagia in chronic diseases where it is prevalent, particularly in neurological pathology such as cerebral palsy (CP), aiming to ensure safe feeding and prioritize proper nutritional management within intervention strategies to meet the complex nutritional needs of these patients [2].

Similar to the adult population, pediatric OPD is associated with a high risk of malnutrition. Malnutrition in this context is multifactorial, influenced by factors such as reduced oral intake, increased nutritional needs, and complications related to the primary condition causing dysphagia. Malnutrition, in turn, can exacerbate functional impairment, delay growth and development, and increase morbidity [3].

Nutritional support is essential in treating malnutrition in pediatric patients with dysphagia. Key strategies include dietary modifications, such as food texturization, and the use of external feeding devices when necessary [5,6,7]. These interventions help ensure adequate nutrient intake and address the unique swallowing challenges faced by these patients, thereby improving their overall health outcomes [8]. All of this, together with periodic monitoring of growth and nutritional status, are essential to ensure the effectiveness of medical interventions and to optimize strategies as needed [9].

Children with neurological impairment, such as cerebral palsy, may experience limited food intake, which, along with the presence of gastrointestinal symptoms, complicates the management of dysphagia itself and leads to a higher risk of malnutrition [10]. Moreover, the nutritional status and severity of dysphagia are closely related to the degree of motor impairment in children with CP [11]. In children with complex chronic pathologies, dysphagia screening is mandatory to avoid underdiagnosis, which also implies the need for adequate assessment and treatment in specialized units.

Dietary modifications and nutritional support are key strategies to address the nutritional needs of patients with dysphagia. These interventions must be individually tailored according to the specific needs of each patient, ensuring adequate nutrition and hydration. This involves adjusting the texture and consistency of foods to facilitate safe swallowing, ranging from pureed to soft solids, and incorporating thickened liquids to prevent aspiration. Nutritional supplements may be necessary to meet caloric and nutrient requirements. In cases where oral intake is insufficient or unsafe, alternative feeding methods such as nasogastric tubes or gastrostomy may be employed. Close monitoring and regular reassessment are essential to adapt the nutritional plan as the patient’s condition evolves, ensuring optimal growth, development, and overall health. Multidisciplinary collaboration, including dietitians, speech-language pathologists, and healthcare providers, is crucial for comprehensive management and support of these patients [12,13].

The combination of dysphagia and malnutrition, along with respiratory complications, can significantly impact the quality of life of patients and their caregivers, limiting daily life activities [14].

Addressing malnutrition in patients with dysphagia, regardless of the underlying cause, is essential for improving health outcomes and enhancing the quality of life of these families and their caregivers. Despite growing awareness of the importance of nutritional management for these patients, significant gaps in knowledge persist regarding the adequacy of diet in terms of specific macro and micronutrients. Therefore, a multidisciplinary and holistic approach is necessary for these patients to perform an adequate diagnosis, treatment, and to plan effective interventions in this vulnerable population.

The objective of our study was to assess caloric intake and contributions of both macro and micronutrients and their possible relation to established nutritional needs, in order to identify potential deficiencies and guide future nutritional interventions.

## 2. Materials and Methods

This was a prospective quasi-experimental study including pediatric patients with swallowing disorders undergoing a specific assessment and treatment program for dysphagia. The study population consisted of patients under 14 years of age diagnosed with dysphagia with impairment of efficacy and/or safety assessed in a specialized pediatric dysphagia clinic during the period from July 2020 to September 2022.

Epidemiological and anthropometric variables were collected. For children under 2 years old, precise scales were used to measure weight, and length was determined accurately. These measurements were taken during the consultation with the patient barefoot and in underwear. Weight, height, and BMI z-scores were calculated using data from Spanish growth charts [15]; dietary intake was also assessed by a nutritionist, using a 3-day dietary record with analysis of both macro and micronutrients. To calculate total energy expenditure (TEE), predictive equations by Schofield for children were used [16,17]. Dietary reference intakes for energy and nutrients were based on values published by the European Food Safety Authority (EFSA). Additionally, patient chronic complexity was classified using the Scale for the Identification of Complex Pediatric Patients (PedCom^®^ Scale) [18].

To establish the severity of dysphagia, the FOIS scale (including pediatric versions) [19,20,21,22] and the score on the EDACS/mini-EDACS scales were used [23,24]. Once all baseline information was collected, patients initiated an individualized speech therapy rehabilitation program. At the end of the follow-up period, outcome variables were obtained considering: (a) anthropometric outcome, according to WHO growth charts in children under 5 years of age or national growth studies in those older than that age (following the recommendations of the European Society for Pediatric Gastroenterology, Hepatology, and Nutrition) [25]; (b) analysis of dietary records, assessing changes in caloric intake, macro and micronutrients.

The maximum tolerated texture in each patient’s oral diet was recorded, utilizing the International Dysphagia Diet Standardisation Initiative (IDDSI) framework to standardize terminology. Patients were stratified into three groups based on their tolerated texture levels: the first group was composed of children who exclusively tolerated significantly modified textures within the IDDSI 0 to IDDSI 4 range, indicating a stringent limitation to these consistencies.; the second group included patients tolerating “minced and moist” foods classified between IDDSI 5 and IDDSI 6; and the third group consisted of children able to tolerate standard diets corresponding to IDDSI 7.

### Statistical Analysis

Variables with a normal distribution were expressed as mean ± standard deviation, and those without it were expressed as median and interquartile range (IQR). The Kolmogorov–Smirnov test was used to assess the normality of the distribution. The chi-square test was used to compare proportions. The t-Student test was used to compare variables with a normal distribution, and the Mann–Whitney U test was used for those without a normal distribution. A logistic regression model was used to estimate propensity scores, with the receipt of enteral nutrition as the dependent variable and age, Z score weight, and Z score height/length, and sex as independent variables. These factors were chosen for their potential influence on both the assignment to enteral nutrition and its outcomes. Matching Procedure (PSM): 1:1 nearest neighbor matching was performed using a caliper width of 0.2 standard deviations of the logit of the propensity score. This strict matching criterion was chosen to ensure the groups were well matched on the baseline covariates, thereby reducing systematic differences. The effectiveness of the matching was assessed by examining the standardized mean differences (SMD) for each covariate before and after matching. Achieving an SMD less than 0.1 for each covariate indicated a successful balance between the treatment and control groups. Following the establishment of well-matched groups, the study proceeded with statistical analyses to evaluate the effect of enteral nutrition on the specified outcome. A *p*-value < 0.05 was considered statistically significant. Data were analyzed using the SPSS^®^ statistical package, version 24.0 for MacOS^®^ (SPSS, Inc., Chicago, IL, USA).

## 3. Results

A total of 117 patients met the inclusion criteria, of which 99 were finally analyzed after obtaining consent from parents. The median follow-up of the patients was 22.3 months (IQR: 16-26.7). During the study period, there were 4 deaths certified, 1 patient was discharged before the minimum expected follow-up time due to evident improvement, and 13 patients were lost to follow-up for various reasons (change of residence to another province or neurological deterioration that justified discontinuation of the rehabilitation program). Table 1 summarizes the main epidemiological and clinical characteristics of the patients included in the study.

At the outset of the follow-up period, the assessment of dietary intake revealed a median intake of 1133.14 kcal/day (IQR 876–1316.5), with 39.8% of participants failing to meet 100% of the estimated total energy expenditure (TEE). In this group of patients, no statistically significant differences were observed according to GMFCS or PedCom^®^, but there were differences in those who had feeding tubes; notably, those with a device had a significantly lower daily caloric intake (*p* = 0.01). Total daily intake was significantly higher in patients with less severe dysphagia according to the FOIS scale adapted for children (*p* = 0.018), with no differences according to the EDACS scale. The mean intake values of both macronutrients and micronutrients and the differences between age groups are presented in Table 2 and Table 3, which also reflect the differences found according to coverage of 80% of the Recommended Dietary Allowance (RDA). Regarding macronutrients, protein intake was higher in the group that did not reach 80% of the RDA, while carbohydrate intake was higher in the group that did meet it.

In assessing dietary adaptations according to the International Dysphagia Diet Standardisation Initiative (IDDSI) framework, it was observed that 52.9% of patients required modified textures corresponding to IDDSI Levels 0–4, suggesting a need for significant dietary adaptation. In contrast, 16.3% of the patients managed with textures associated with IDDSI Levels 5–6, indicating a capability for more advanced textures such as minced and moist foods, while 30.8% were able to handle regular textures, corresponding to IDDSI Level 7. Contingency table analysis revealed statistically significant differences in IDDSI levels across the age stratifications (*p* < 0.001), suggesting an age-related variation in the capability to tolerate different food textures.

Of the patients supported with enteral nutrition (42.7%), 26% received enteral nutrition contributions that constituted more than 80% of the total energy intake. Additionally, 41.3% of this subgroup utilized an external feeding device. Table 4 outlines the variations in micro and macronutrient intake between patients with and without enteral supplementation, following a propensity score matching (PSM) analysis. Children in the nutritional support group exhibited significantly higher percentages of caloric requirement coverage (*p* = 0.023).

In addressing dysphagia, the speech therapy regimen incorporated both compensatory and direct rehabilitative approaches. Compensatory strategies facilitated patients’ adjustments to their swallowing challenges, including dietary modifications and alterations to feeding routines. Rehabilitative efforts were directed at rectifying the underlying dysfunctions of the swallowing process. These approaches were individualized, considering factors such as the type of dysphagia, patient age, and levels of cooperation. Particularly in patients with neurological disorders, a synthesis of compensatory and rehabilitative methods was generally adopted. The outcomes, which detail the relationship between the severity of swallowing impairments and the extent of dietary adaptation required, are presented in Table 5. Notably, those with more severe swallowing safety concerns necessitated more substantial adjustments in dietary volume, texture, and viscosity compared to those with milder EDACS safety scores (*p* = 0.032).

Another outcome measure was dietary analysis after at least one year of follow-up. The results revealed a statistically significant difference in caloric intake before (1133 ± 382.7 kcal) and after treatment (1374.7 ± 407.7 kcal), *p* < 0.001. However, when evaluating intake based on estimated TEE, the results did not show a significant difference before and after follow-up; there were average values of energy coverage according to daily requirements that increased after treatment (95.8 ± 26.2% vs. 119.8 ± 41.3%, *p* < 0.001). A total of 31% of patients who did not reach 100% of theoretical requirements according to TEE achieved it after the rehabilitation period (*p* = 0.017). There was a major increase in the protein and carbohydrate and also fats intake per day, which was statistically significant (*p* ≤ 0.001), reflecting an overall enhancement in macronutrient intake. The proportions of these macronutrients relative to the total caloric value were maintained, indicating a sustained balance in dietary composition; furthermore, micronutrient intake remained constant, with no significant differences observed. There was also a significant improvement in the maximum texture levels that patients could tolerate from the first encounter with IDDSI to the last follow-up. Hence, the Z-score of −3.8 is statistically significant (*p* < 0.001), within the 95% CI.

## 4. Discussion

This study enhances understanding of nutritional management in children with dysphagia, highlighting deficiencies in energy and micronutrient intake, as well as the benefits of individualized dietary and therapeutic interventions. It underscores the complexity of nutritional management in this demographic, emphasizing the need for specific nutritional strategies for complex chronic conditions such as CP aligning with previous research [26,27,28].

It was observed that a considerable proportion of children with chronic conditions and oropharyngeal dysphagia do not meet the estimated total energy expenditure (TEE) requirements. This concurs with Cieri M et al. [26] who documented variations in energy and nutritional intake in children with CP, correlating nutritional status with the severity of motor impairment. This parallel underscores a common nutritional concern among vulnerable pediatric populations, highlighting the importance of implementing evidence-based nutritional interventions.

The high prevalence of neurological or complex chronic conditions among our sample supports the documented epidemiology of dysphagia in children, confirming the increased risk of malnutrition identified by Penagini et al. [29] due to feeding difficulties in children with neurological disabilities. The association with the severity of motor impairment reinforces the need for a comprehensive approach to nutritional status. A subsequent series of children with CP [10], where 82.5% exhibited symptoms of dysphagia, reflects the significant impact this has on daily energy intake as well as macronutrients and fluids. Children with dysphagia consumed less energy, carbohydrates, and fluids compared to those without dysphagia, highlighting the direct relationship between dysphagia and reduced food intake, which can contribute to malnutrition.

Ruiz Brunner et al. [30] in Argentina demonstrated that the risk of severe malnutrition increases with the degree of motor impairment, and the discrepancies between subjective and objective perceptions of nutritional status, noted by Fogarasi A et al. [31], underscore the importance of rigorous nutritional assessment in these populations. This, along with our findings, reinforces the value of a multidisciplinary management of pediatric dysphagia, integrating appropriate nutritional strategies.

Nutrition significantly affects the status and overall development of children with dysphagia, particularly in the presence of chronic diseases such as CP. The literature highlights the complexity of nutrition in these patients, insisting on the personalization of feeding strategies [10,11,25,28,31].

More than half of the children in follow-up required modified textures according to the IDDSI, demonstrating the need to improve dietary formulations to meet nutritional requirements, as the nutritional density of adapted foods is often insufficient. Our research shows post-intervention improvements in caloric intake and tolerated texture levels, suggesting the effectiveness of rehabilitative therapies. The improvement in tolerance to various food textures highlights the relevance of designing rehabilitation programs that not only enhance functionality but also nutritional status, positively impacting growth and development.

Özder F et al. [27] analyzed the relationship between masticatory ability and nutrient intake, observing that caloric and macronutrient intake significantly depends on masticatory skills. This underscores the importance of an early multidisciplinary assessment and rehabilitation to improve dietary tolerance and ensure adequate and safe nutritional intake.

Our data show that enteral nutrition not only helps to meet daily energy requirements but also improves the profile of essential micronutrient intake, achieving more complete and balanced nutrition. Carman K et al. [28] revealed nutritional deficiencies in children with CP, highlighting differences according to the feeding method of the children. The authors find that patients receiving EN via gastrostomy and certain EN solutions can influence the levels of certain micronutrients, such as increased selenium and decreased zinc. It would therefore be necessary to select enteral formulations to adjust not only macronutrient requirements but also essential micronutrients. Vernon-Roberts et al. [32] studied changes in growth, body composition, and micronutrient status in patients with severe disabilities who began feeding via gastrostomy with a low-energy density formula, pointing out the importance of having pediatric enteral formulations that meet macro- and micronutrient requirements, regardless of the proportion that the formula represents of the total daily intake.

### Limitations and Future Directions for Research

The inherent limitations of the quasi-experimental and single-center design, along with the potential loss of follow-up, suggest caution in generalizing the results. Additionally, the authors should include in the limitations of their study that the sample size was small, highlighting the need for future studies with larger sample sizes as well as the performance of randomized controlled clinical trials. Future research should focus on evaluating the long-term effects of nutritional and therapeutic interventions in broader and less heterogeneous populations to strengthen knowledge in the management of pediatric dysphagia.

## 5. Conclusions

A meticulous and personalized nutritional approach in managing children with dysphagia is crucial, especially in patients with complex chronic conditions like cerebral palsy. The findings of this research underscore the need to provide dietary formulations that address not only caloric requirements but also the supply of essential micronutrients to ensure optimal nutritional status for proper development. The introduction of adapted feeding strategies and the use of enteral nutrition are critical to improving both the quality of life and health outcomes in this vulnerable population. Multidisciplinary and individualized approaches are essential to ensure improvements in the nutritional treatment of children with special clinical needs.

## Figures and Tables

**Table 1 nutrients-16-01590-t001:** Epidemiological characteristics of the patients (*n* = 99).

		*n* (%)
Gender (Males)		54 (54.5)
Age years (median, IQR)		2.71 (1.43:4.83)
Disease group		
	Congenital brain damage	53 (53.5)
	Acquired brain damage	13 (13.1)
	Neurodegenerative disease	3 (3)
	Neurodevelopmental disorder	6 (6.1)
	Craniofacial anomaly	11 (11.1)
	Cardiorespiratory pathology	2 (2)
	Functional/Maturational	11 (11.1)
GMFCS		
	I	44 (44.4)
	II	11 (11.1)
	III	10 (10.1)
	IV	13 (13.1)
	V	20 (20.2)
PedCOM complexity level		
	No complexity	41 (41.4)
	Low complexity	29 (29.3)
	Moderate complexity	22 (22.2)
	High complexity	3 (3)
Recurrent respiratory processes		30 (30.3)
	Aspiration pneumonia	1 (1)
Digestive comorbidity		46 (46.4)
	GERD	24 (24.2)
	Constipation	7 (7)
	GERD and constipation	15 (15.1)
Presence of external feeding device (Y/N)		29 (29.3)
	NGT/NYT	23 (23.2)
	Gastrostomy	6 (6)
Referral		
	Primary care	6 (6.1)
	Hospitalized care	93 (93.9)
	Neuropediatric	30 (30.3)
	Palliative care/Complex Chronic	13 (13.1)
	Gastroenterology	19 (19.1)
	Neonatology	8 (8)
	Pneumologist	5 (5)
	Others	18 (18.2)
Reason for referral		
	Evaluate swallowing safety	29 (293)
	Cough with intake	12 (12.1)
	Difficulty expanding oral diet	37 (37.3)
	SDR incoordination	14 (14.1)
	Suspicion of chronic aspirative syndrome	2 (2)
	Others	5 (5.1)
Anthropometry t0 (mean ± SD)		
	z-score Weight	−1.54 ± 1.75
	z-score Height	−1.62 ± 1.92
	z-score BMI	−0.91 ± 1.86

Abbreviations: M: male; F: female; IQR: interquartile range; GMFCS: Gross Motor Function Classification System; PedCOM: Scale for the identification of the complex chronic pediatric patient; Y/N: yes/no; NGT: nasogastric tube; TPN: transpyloric tube; PEG: percutaneous endoscopic gastrostomy; SDR: suck-swallow-breathe; t0: initial assessment; BMI: body mass index; SD: standard deviation.

**Table 2 nutrients-16-01590-t002:** Daily averages of caloric intake, macro and micronutrients stratified by age.

Intake		Mean (SD)	Range	*p*
Energy (kcal/day)		1109.42 (413.72)	[848.19–1.314.6]	
	<1 year	627.04 (195)	[437.1–751.5]	<0.001
	1–7 years	1.174 (342.57)	[967.45–1321.465]	<0.001
	>7 years	1457.95 (338.48)	[1.258–1.612.78]	<0.005
Proteins (g/day)		39.15 (19.98)	[28.03–50.85]	
	<1 year	17.57 (7.23)	[12.57–24.29]	<0.001
	1–7 years	41.41 (11.74)	[32.64–50.79]	<0.001
	>7 years	53.65 (10.02)	[47.35–60.09]	<0.005
Proteins (%TCI)		17.33 (13.83)	[11.77–16.19]	
	<1 year	10.9 (2.45)	[8.66–13.08]	<0.001
	1–7 years	14.39 (3.38)	[11.90–16.67]	<0.001
	>7 years	14.9 (1.9)	[13.21–16.84]	0.005
Fat (g/day)		42.72 (19.48)	[31.55–50.09]	
	<1 year	25.93 (8.97)	[19.48–34.21]	0.004
	1–7 years	43.76 (8.33)	[34.18–50.74]	<0.005
	>7 years	58 (20.14)	[42.28–73.24]	<0.005
Fat (%TCI)		42.26 (77.43)	[29.02–39.58]	
	<1 year	38.51 (11.65)	[26.41–48.39]	<0.05
	1–7 years	33.07 (7.31)	[28.87–38.01]	<0.005
	>7 years	35.28 (5.66)	[29.31–40.79]	0.47
Carbohydrates (g/day)		145.62 (54.08)	[108.41–179.67]	
	<1 year	80.41 (32.97)	[62.18–102.62]	<0.001
	1–7 years	154.14 (48.86)	[123.62–182.61]	<0.001
	>7 years	180.32 (5.65)	[161.22–189.73]	0.08
Carbohydrates (%TCI)		67.33 (149.45)	[46.17–56.74]	
	<1 year	50.35 (9.53)	[43.01–59.66]	0.73
	1–7 years	52.66 (8.33)	[46.89–56.82]	1
	>7 years	49.82 (5.09)	[46.05–54.81]	1
Fiber (g/day)		6.54 (6.75)	[2.39–7.37]	
	<1 year	2.09 (2.13)	[0–3.51]	<0.05
	1–7 years	5.55 (7.14)	[3.28–9.62]	1
	>7 years	6.53 (6.37)	[2.02–9.41]	0.3

SD: standard deviation; TCI: total caloric intake.

**Table 3 nutrients-16-01590-t003:** Characterization of the sample according to the extent of intake coverage above or not above 80% of the recommendations by age.

Age	Coverage ≥80% RDI	Coverage <80% RDI	
	*n* (%)	*n* (%)	*p*
<1 year	13 (17.3%)	4 (12.5)	0.01
1–7 years	57 (76%)	19 (59.4)	
>7 years	5 (21.6%)	9 (28.1)	
**GMFCS**	***n* (%)**		** *p* **
I–III	50 (70.4%)	23 (74.2)	0.698
IV–V	21 (29.6%)	8 (25.8)	
**PedCom^®^**	***n* (%)**	***n* (%)**	** *p* **
No complexity	38 (55.1)	13 (43.3)	0.14
Low complexity	20 (29)	6 (20)	
Moderate complexity	5 (7.2)	4 (13.3)	
High complexity	6 (8.7)	7 (23.3)	
**EDACS**	***n* (%)**	***n* (%)**	** *p* **
Levels 1–3	61 (88.4)	26 (86.7)	0.81
Levels 4–5	8 (11.6)	4 (13.3)	
**External feeding device**	***n* (%)**	***n* (%)**	** *p* **
None	54 (74)	23 (74.2)	0.98
NGT/Gastrostomy	19 (26)	8 (25.8)	
**Macronutrients**	**Median [IQR]**		** *p* **
Proteins (%, g/day)	13.81; 41.84 [30.48–51.44]	14.25; 34.24 [22.94–47.4]	0.014
Carbohydrates (%, g/day)	52.59; 160.19 [130.61–190.63]	50.95; 117.55 [86.48–145.1]	<0.001
Fat (%, g/day)	33.65; 45.65 [34.25–53.03]	34.79; 35.47 [23.6–45.74	0.004
**Micronutrients**	**Median [IQR]**		
Calcium (mg/day)	716.33 [465.84–934.63]	591.35 [392.39–747.03]	0.133
Phosphorus (mg/day)	486.95 [326.89–585.64]	361.27 [170.71–488.88]	0.009
Magnesium (mg/day)	120.12 [72.25–154.95]	95.87 [52.76–134.32]	0.054
Iron (mg/day)	8.26 [4.80–10.25]	5.96 [3.27–7.53]	0.029
Vitamin D (mcg/day)	8.05 [1.93–10.87]	4.99 [0.97–7.03]	0.13

RDI: recommended daily intake; IQR: interquartile range; GMFCS: Gross Motor Function Classification System; PedCom^®^: scale for the identification of the complex chronic pediatric patient; EDACS: Eating and Drinking Ability Clasification System; NGT: nasogastric tube.

**Table 4 nutrients-16-01590-t004:** Daily averages of caloric intake, macro- and micronutrients stratified by feeding with enteral nutrition (after PSM).

	Non-Enteral Nutrition Support (*n* = 17)	Enteral Nutrition Support (*n* = 28)	*p*
Age, median (IQR)	3.4 (2.1–6.4)	2.8 (1.3–4.2)	0.223
Sex (females), n %	8 (47.1)	10 (35.7)	0.329
Weight (z score)	−2.2 (−2.7–−1.6)	−2.4 (−3.3–−1.9)	0.157
Height/Length (z score)	−2.3 (−3.9–−1.7)	−2.1 (−2.9–−1.3)	0.482
TEE (kcal), median [IQR]	1070 [774–1159]	977 [671–1149]	0.287
DER, median [IQR]	1252 [1252–1573]	1252 [1000–1333]	0.112
Total daily energy, median [IQR]	1102 [915–1161]	1100 [844–1355]	0.607
DER Covered (%), median [IQR]	78 [62–86]	90 [75–106]	0.037
% Energy from TEE, median [IQR]	101 [85–120]	117 [92–155]	0.083
Energy/kg BW, median [IQR]	83 [61–101]	95 [75–127]	0.042
Macronutrients	Median [IQR]	Median [IQR]	*p*
Proteins (%, g/day)	15.1; 40.6 [29.8–51.1]	13.6; 37.1 [22.9–48.9]	0.574 0.198
Carbohydrates (%, g/day)	47.1; 128.9 [116.1–143.5]	50.1; 137.4 [85.6–178.5]	0.233 0.292
Fat (%, g/day)	37.1; 44.1 [28.0–50.1]	35.4; 42.5 [30.4–51.4]	0.870 0.963
Fiber (g/day)	3.1 [1.6–6.9]	4.9 [2.2–10.7]	0.210
% RDI fiber covered	24.9 [13.7–50.2]	41.4 [22.8–106.7]	0.204
Micronutrients	Median [IQR]	Median [IQR]	*p*
Calcium (mg/day)	697.9 [415.2–840.9]	602 [466.1–965.2]	0.640
% RDI calcium covered	94.9 [59.2–127.5]	102.9 [77.5–158.8]	0.292
Phosphorus (mg/day)	338.1 [180.5–509.1]	486.8 [343.1–674.1]	0.033
% RDI phosphorus covered	61.5 [37.7–104.3]	106.3 [74.8–130.9]	0.004
Magnesium (mg/day)	100.5 [70.2–130.5]	123.2 [71.7–170.3]	0.153
% RDI magnesium covered	82.3 [51.4–105.6]	121.3 [74.9–188.9]	0.029
Iron (mg/day)	3.7 [2.5–5.1]	8.7 [5.3–11.0]	0.0001
% RDI iron covered	45.1 [29.6–63.5]	110.9 [66.9–156.7]	0.0001
Zinc (mg/day)	2.3 [0.8–4.3]	5.4 [4.2–8.8]	0.002
% RDI zinc covered	39.7 [16.2–71.8]	115.9 [90.4–197.7]	0.0001
Vitamin D (mcg/day)	1.7 [0.4–4.4]	7.7 [4.2–12.8]	0.002
% RDI vitamin D covered	26.2 [5.4–71.4]	115.0 [61.0–167.3]	0.002

IQR: interquartile range; TEE: total energy expenditure; DER: daily energy requirement; BW: body weight; RDI: recommended daily intake.

**Table 5 nutrients-16-01590-t005:** Strategies used during speech therapy rehabilitation according to the underlying pathology, EDACS feeding safety scale, and the presence or absence of an external feeding device.

	Neurologic Disease			GMFCS			EDACS			External Feeding Device		
Strategies	Yes	No	*p*	I–III	IV–V	*p*	1–3	4–5	*p*	Yes	No	*p*
Compensatory	11 (13.9%)	6 (30%)	0.016	13 (20.3%)	4 (11.8%)	<0.001	14 (16.5%)	3 (21.4%)	0.032	10 (14.3%)	7 (24.1%)	0.184
Direct Rehabilitation	20 (25.3%)	9 (45%)		28 (43.8%)	1 (2.9%)		29 (34.1%)	0 (0%)		24 (34.3%)	5 (17.2%)	
Both	48 (60.8%)	5 (25%)		23 (35.9%)	29 (85.3%)		42 (49.4%)	11 (78.6%)		36 (51.4%)	17 (58.6%)	

## Data Availability

The data that support the findings of this study are not openly available due to reasons of sensitivity and are available from the corresponding author upon reasonable request.

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
