# Peer review of "Nutritional Issues in Children with Dysphagia"

_nutrients, 2024, doi:10.3390/nu16111590_

Round 1

Reviewer 1 Report

Comments and Suggestions for Authors

This is an interesting reserach with adequate novelt and quality focused on the nutritiona issues in childern with dysphagia. However, several points should be adrerssed before reconsideratrion.

- In the abstract section, in line 15, the autors reported that " This is prospective quasi-experimental study". The word "quasi" is not broadly known. Is this any others scientific orology that you can used indead of it?

- In the abstract secrion, il lines 15-16, the authors reported as sample size:  117 children under 14 years old. The authors  should more be specified reporting the range of the age. Is this study include a more specidic age group of children, e.g., aged 6-9, 9-12, 12-15 years?

- The 1st paragraph of the introduction section needs more relevant reference, whereas the second paragraph did noy includre any relevant refereces. 

- The authors should report and describ the diagnostic criteria of dysphagia, including a relevan reference.

In addirion, the authors should report the potential risk factors of dysphagia and the pathophysiologic mechanisms governing dysphagia.

- Again, 3rd paragraph did not include any relevant references. This paragraph should include at least 293 relevant referenves.

- The authors report in lines 49-51 that " alongside dietary modifications, including food texturization and/or the incorporation of external feeding devices when necessary, are key strategies in managing these patients." This statement is quite complex and needs rephasing. Moreover, this statement should be accompinied by relevant reference.

- In the last sentence of the 4rd paragraph (lines 58-69), a reference is need to support this statement.

- More information concerning the topic of the sentence in lines 62-64 " These interventions must be individually tailored according to the specific needs of each patient, ensuring adequate nutrition and  hydration (6)."

- The paragraph in linse 66-68 needs more relevant text and the authors should report additional studies with their relevant references.

- In the begining of the Materials and methods section the authors reported that thi is a oprospective quasi-experimental study.e word "quasi" is not broadly known. Is this any others scientific orology that you can used indead of it?

- In line 86. the authors should include the range of the enrlolled children, as well as mean+/-Standard deviation (if age is a normally distrubuted variable) or the median /Interquartile range (if the age is no normally distibution variable).

- In line 88, the authors should reportic whether thw anthropometric parametes (such as body mass weight and height or waist circumference) are self reported data or measured data. If the anthropometric parameters are slef-reported data then recall bias may existh and this statement should be invluded as a limitation of the study at the end of the discussion section.

- In line 99, the authors should state why they finally analyzed 99 children and not the initially enrolled 117 children.

- In Table 1, the symbol % should be included into each bracket.

- The paragraph in lines 247-250 should be increased by simultaneously adding relevant references.

- The authors should include in the limitations of their study that tha sample size was small, highlighting the need for future studies including larger sample size as well the performance of randomized controlled clinical trials.

- Some more recent references publidhed in the last 2-3 years should be included.

Comments on the Quality of English Language

Moderate editing of English language required

Author Response

Reviewer #1 comments:

Comments and suggestions for authors

This is an interesting reserach with adequate novelt and quality focused on the nutritiona issues in childern with dysphagia. However, several points should be adrerssed before reconsideratrion.

Answer: Thanks. The comment is highly appreciated.

Comment 1: In the abstract section, in line 15, the autors reported that " This is prospective quasi-experimental study". The word "quasi" is not broadly known. Is this any others scientific orology that you can used indead of it?

Answer 1: A quasi-experimental study is an empirical interventional study used to estimate the causal impact of an intervention on a target population without random assignment. Quasi-experimental research shares similarities with traditional experimental design or randomized controlled trials, but it specifically lacks the element of random assignment to treatment or control groups. Instead, quasi-experimental designs typically allow the researcher to control the assignment to the treatment condition using some criterion other than random assignment. This is exactly what we did, so we’ll keep it.

Comment 2: In the abstract secrion, il lines 15-16, the authors reported as sample size:  117 children under 14 years old. The authors  should more be specified reporting the range of the age. Is this study include a more specidic age group of children, e.g., aged 6-9, 9-12, 12-15 years?

Answer 2:  Added.

Comment 3: The 1st paragraph of the introduction section needs more relevant reference, whereas the second paragraph did noy includre any relevant refereces. Done

Answer 3:

Comment 4: The authors should report and describ the diagnostic criteria of dysphagia, including a relevan reference.

In addirion, the authors should report the potential risk factors of dysphagia and the pathophysiologic mechanisms governing dysphagia.

Answer 4: Thank you so much for the comment. Done

Comment 5: Again, 3rd paragraph did not include any relevant references. This paragraph should include at least 293 relevant referenves.

Answer 5: Thank you so much for the comment. Done

Comment 6: The authors report in lines 49-51 that " alongside dietary modifications, including food texturization and/or the incorporation of external feeding devices when necessary, are key strategies in managing these patients." This statement is quite complex and needs rephasing. Moreover, this statement should be accompinied by relevant reference.

Answer 6: Thank you so much for the comment. Done

Comment 7: In the last sentence of the 4rd paragraph (lines 58-69), a reference is need to support this statement.

Answer 7: Thank you so much for the comment. Done

Comment 8: More information concerning the topic of the sentence in lines 62-64 " These interventions must be individually tailored according to the specific needs of each patient, ensuring adequate nutrition and  hydration (6)."

Answer 8: Thank you so much for the comment. Done

Comment 9: The paragraph in linse 66-68 needs more relevant text and the authors should report additional studies with their relevant references

Answer 9: Thank you so much for the comment. Done

Comment 10: In the begining of the Materials and methods section the authors reported that thi is a oprospective quasi-experimental study.e word "quasi" is not broadly known. Is this any others scientific orology that you can used indead of it?

Answer 10: A quasi-experimental study is an empirical interventional study used to estimate the causal impact of an intervention on a target population without random assignment. Quasi-experimental research shares similarities with traditional experimental design or randomized controlled trials, but it specifically lacks the element of random assignment to treatment or control groups. Instead, quasi-experimental designs typically allow the researcher to control the assignment to the treatment condition using some criterion other than random assignment. This is exactly what we did, so we’ll keep it.

Comment 11: In line 86. the authors should include the range of the enrlolled children, as well as mean+/-Standard deviation (if age is a normally distrubuted variable) or the median /Interquartile range (if the age is no normally distibution variable).

Answer 11: Thank you for the comment. These data are shown on table 1.

Comment 12: In line 88, the authors should reportic whether thw anthropometric parametes (such as body mass weight and height or waist circumference) are self reported data or measured data. If the anthropometric parameters are slef-reported data then recall bias may existh and this statement should be invluded as a limitation of the study at the end of the discussion section.

Answer 12: Thank you for the comment. Done

Comment 13: In line 99, the authors should state why they finally analyzed 99 children and not the initially enrolled 117 children.

Answer 13: Thank you for the comment. Done

Comment 14: In Table 1, the symbol % should be included into each bracket.

Answer 14: In our opinion, this is not necessary because it is already explained at the top of the table.

Comment 15: The paragraph in lines 247-250 should be increased by simultaneously adding relevant references.

Answer 15: Done

Comment 16: The authors should include in the limitations of their study that tha sample size was small, highlighting the need for future studies including larger sample size as well the performance of randomized controlled clinical trials.

Answer 16: Done.

Comment 17: Some more recent references publidhed in the last 2-3 years should be included.

Answer 17: Thank you for the comment. Done.

Reviewer 2 Report

Comments and Suggestions for Authors

Dear authors,

I have now completed the review of the manuscript titled "Nutritional issues in children with dysphagia."

The manuscript is interesting and, in general, fairly well-written.

I have some suggestions to further improve the quality of the manuscript.

I would like to suggest that the authors address these limitations in the article, either by discussing them in the limitations section or, where feasible, by making the appropriate revisions:

1. While explaining backgrounds, authors should mention latest GBDs on upper digestive systems. For example, Global prevalence of functional dyspepsia according to Rome criteria, 1990-2020: A systematic review and meta-analysis.

2. The study attempts to control for various factors using propensity score matching, but residual confounding is always a risk in observational studies. Including more covariates or utilizing more sophisticated statistical methods could help mitigate this issue. For example, try to get some covariates from Associations between Delayed Introduction of Complementary Foods and Childhood Health Consequences in Exclusively Breastfed Children OR Association between complementary food introduction before age 4 months and body mass index at age 5-7 years: a retrospective population-based longitudinal cohort study.

3. The relatively small sample size (117 initial participants, with 99 completing the study) and the potential biases introduced by the loss of follow-up participants (17.1%) could affect the study's robustness. Increasing the sample size and ensuring more rigorous follow-up could address these issues. For example, authors may rebuttal Relationship between feeding to sleep during infancy and subsequent childhood disease burden. 

4. The study emphasizes short-term improvements in caloric intake and dietary tolerance but does not provide sufficient data on long-term health and developmental outcomes. Longitudinal studies are needed to understand the sustained impact of nutritional interventions. For example, Rotavirus-Associated Hospitalization in Children With Subsequent Autoimmune Disease can be reffered.

Thank you for your valuable contributions to our field of research. I look forward to receiving the revised manuscript.

Author Response

Reviewer #2 comments:

Comments and suggestions for authors:

I have now completed the review of the manuscript titled "Nutritional issues in children with dysphagia."

The manuscript is interesting and, in general, fairly well-written.

I have some suggestions to further improve the quality of the manuscript.

I would like to suggest that the authors address these limitations in the article, either by discussing them in the limitations section or, where feasible, by making the appropriate revisions:

Comment 1: While explaining backgrounds, authors should mention latest GBDs on upper digestive systems. For example, Global prevalence of functional dyspepsia according to Rome criteria, 1990-2020: A systematic review and meta-analysis.

Answer 1:  We regret to contradict the reviewer, but we do not fully understand the relation of the proposed citation to the topic of our article. Could you please clarify?

Comment 2: The study attempts to control for various factors using propensity score matching, but residual confounding is always a risk in observational studies. Including more covariates or utilizing more sophisticated statistical methods could help mitigate this issue. For example, try to get some covariates from Associations between Delayed Introduction of Complementary Foods and Childhood Health Consequences in Exclusively Breastfed Children OR Association between complementary food introduction before age 4 months and body mass index at age 5-7 years: a retrospective population-based longitudinal cohort study.

Answer 2: Thank you very much to the reviewer for their comment. Unfortunately, we cannot comply with the request because we did not collect those variables. We will consider this suggestion for future studies. Thank you.

Comment 3:  The relatively small sample size (117 initial participants, with 99 completing the study) and the potential biases introduced by the loss of follow-up participants (17.1%) could affect the study's robustness. Increasing the sample size and ensuring more rigorous follow-up could address these issues. For example, authors may rebuttal Relationship between feeding to sleep during infancy and subsequent childhood disease burden.

Answer 3:  We appreciate the reviewer's excellent comment. We have added the sample size to our limitations. Unfortunately, we do not have those data collected and therefore cannot perform the requested analysis. Thank you very much.

Comment 4: The study emphasizes short-term improvements in caloric intake and dietary tolerance but does not provide sufficient data on long-term health and developmental outcomes. Longitudinal studies are needed to understand the sustained impact of nutritional interventions. For example, Rotavirus-Associated Hospitalization in Children With Subsequent Autoimmune Disease can be reffered.

Answer 4: Thank you very much for the comment. This is reflected in our study as a limitation. Thank you.

Round 2

Reviewer 1 Report

Comments and Suggestions for Authors

The authors have significantly improved their manuscript after the revision process.

Comments on the Quality of English Language

Minor editing of English language required